# The Nitric Oxide (NO) Donor Molsidomine Counteract Social Withdrawal and Cognition Deficits Induced by Blockade of the NMDA Receptor in the Rat

**DOI:** 10.3390/ijms24076866

**Published:** 2023-04-06

**Authors:** Lamprini Katsanou, Evangelia Fragkiadaki, Sotirios Kampouris, Anastasia Konstanta, Aikaterini Vontzou, Nikolaos Pitsikas

**Affiliations:** Department of Pharmacology, Faculty of Medicine, School of Health Sciences, University of Thessaly, Biopolis, Panepistimiou 3, 415-00 Larissa, Greece

**Keywords:** nitric oxide, molsidomine, clozapine, ketamine, schizophrenia, rat

## Abstract

The deficiency of the gaseous molecule nitric oxide (NO) seems to be critically involved in the pathogenesis of schizophrenia. Thus, molecules that can normalize NO levels, as are NO donors, might be of utility for the medication of this psychiatric disease. The aim of the present study was to detect the ability of the NO donor molsidomine to reduce schizophrenia-like impairments produced by the blockade of the N-methyl-D-aspartate (NMDA) receptor in rats. Molsidomine’s ability to attenuate social withdrawal and spatial recognition memory deficits induced by the NMDA receptor antagonist ketamine were assessed using the social interaction and the object location test, respectively. Further, the efficacy of the combination of sub-effective doses of molsidomine with sub-effective doses of the atypical antipsychotic clozapine in alleviating non-spatial recognition memory deficits was evaluated utilizing the object recognition task. Molsidomine (2 and 4 mg/kg) attenuated social withdrawal and spatial recognition memory deficits induced by ketamine. Co-administration of inactive doses of molsidomine (1 mg/kg) and clozapine (0.1 mg/kg) counteracted delay-dependent and ketamine-induced non-spatial recognition memory deficits. The current findings suggest that molsidomine is sensitive to glutamate hypofunction since it attenuated behavioral impairments in animal models mimicking the negative symptoms and cognitive deficits of schizophrenia. Additionally, the present results support the potential of molsidomine as an adjunctive drug for the therapy of schizophrenia.

## 1. Introduction

Schizophrenia is a complex, devastating psychiatric disorder. Epidemiological evidence suggests that schizophrenia affects up to 1% of the world’s population. Schizophrenia disrupts social, occupational, and individual functioning and impairs the quality of life of patients. This disease is usually manifested in former youth or early maturity. Schizophrenia patients display grave psychotic symptoms, which can be divided into three distinct groups: positive symptoms (hallucinations, delusions, thought disorder, hyperactivity, disorganized speech, and bizarre behaviors), negative symptoms (flat expressions, social withdrawal, anhedonia, alogia, and avolition) and cognitive impairments (disturbances in attention, executive functioning, and memory) [1].

Consistent experimental evidence suggests that hypoactivity of the glutamatergic system is associated with schizophrenia. Abnormal glutamatergic transmission induces secondary dopaminergic (DAergic) dysfunction in the striatum and prefrontal cortex (PFC). It has been shown that the pharmacological blockade of the N-methyl-*D*-aspartate (NMDA) receptor elicits negative symptoms and cognitive impairments that were not relieved by currently used antipsychotics [2]. NMDA receptor antagonists compounds [e.g., MK-801, phencyclidine (PCP), and ketamine] are able to induce behavioral symptoms mimicking positive as well negative symptoms and cognitive deficits of schizophrenia in healthy subjects [3,4], exacerbate symptomatology in schizophrenia patients [5,6], and induce schizophrenia-like symptoms, including negative symptoms and cognitive deficits, in rodents [7,8,9,10]. Currently used antipsychotics have shown some utility exclusively in the relief of the positive symptoms of schizophrenia, while 30% of patients do not respond to any pharmacological intervention. This suggests an urgent need for novel neuroleptics with a higher therapeutic profile that might be able to alleviate negative and cognitive symptoms clusters and provide a benefit for treatment-resistant schizophrenia patients [11].

Among the different alternative approaches for the therapy of schizophrenia, the involvement of the gas nitric oxide (NO) modulators as potential anti-schizophrenia agents has lately been suggested. NO is a soluble, short-lived, and freely diffusible gas. It is an important intra- and inter-cellular messenger in the brain [12]. The implication of NO in schizophrenia is well-documented [for review, please see 13]. The underproduction of NO seems to be related to schizophrenia [13]. Thus, normalization of NO brain concentrations in schizophrenia patients might be beneficial in this context. In line with the above, molecules that enhance NO production, as are NO donors, might represent a novel therapeutic tool for the treatment of this psychiatric disease. This view has been corroborated by preclinical findings. Specifically, the efficacy of different NO donors, including sodium nitroprusside (SNP) and molsidomine, on psychotomimetic effects and cognition impairments produced by blockade of glutamate or by dysregulation of DAergic neurotransmission has been evidenced (for review, please see [14]).

Despite the encouraging preclinical findings, research conducted on schizophrenia patients did not yield the same conclusions. The efficacy of the NO donor SNP was evidenced only in distinct categories of schizophrenia patients, such as those who are young, no smokers, and in the early stage of the illness. By contrast, SNP failed to attenuate schizophrenia symptoms in older cigarette smokers and chronic patients [15,16,17,18,19,20]. In this regard, it is important to underline that administration of SNP is often associated with important adverse effects such as methemoglobinemia [21] and sedation [22]. Finally, SNP’s pharmacological action is characterized by a narrow therapeutic window [23].

Molsidomine is a NO-releasing agent, which, like SNP, is used for the treatment of different forms of coronary diseases. Molsidomine is a prodrug and is quickly metabolized to its active metabolite 3-morpholinosydnonimine (SIN-1) [24]. Interestingly, molsidomine does not cause methemoglobinemia and displays a longer duration of action (2 h) [25] with respect to that expressed by SNP (4 min) [26]. Previous preliminary research has shown that molsidomine counteracted non-spatial recognition memory deficits, stereotypies, and ataxia but not hypermotility induced by the NMDA receptor antagonist MK-801 in the rat [27]. At the moment, it is not yet elucidated whether molsidomine can attenuate social withdrawal (an animal model of negative deficits of schizophrenia) [9] and disruption of spatial recognition memory caused by the blockade of the glutamatergic system. Interestingly, recognition memory is a cognitive domain severely disturbed in schizophrenia patients [28,29].

Based on the above evidence, the purpose of the present study was first to examine in the rat the ability of molsidomine to alleviate social isolation induced by the NMDA receptor antagonist ketamine. To this end, the social interaction test (SIT) was used [9]. Subsequently, the efficiency of molsidomine to counteract spatial recognition memory deficits caused by ketamine was studied utilizing the object location task (OLT) [30]. We hypothesize that molsidomine will attenuate social isolation and spatial recognition memory deficits induced by ketamine. If molsidomine expresses an efficacy in these preclinical glutamatergic models of schizophrenia, this might be of interest since, until now, there has been no relief for the negative symptoms and cognitive impairments of schizophrenia [11]. Additionally, we intended to evaluate whether the combination of sub-effective doses of molsidomine with sub-effective doses of clozapine was able to alleviate delay-dependent and ketamine-induced non-spatial recognition memory deficits. Clozapine is an atypical antipsychotic and has shown some utility for treatment-resistant schizophrenia patients [1]. An advantage of the combined administration is that it offers the prospective of a therapeutic benefit and the potential to lower doses of chemicals that may cause undesired side effects. In this context, it has been shown that the administration of sub-effective doses of SNP with sub-effective doses of different atypical antipsychotics enhanced animals’ performance in models of positive symptoms and attentional deficits of schizophrenia [31,32,33]. For these latter studies, we utilized the object recognition task (ORT) [34].

## 2. Results

### 2.1. Experiment 1: Effects of Molsidomine on Ketamine-Induced Social Withdrawal Assessed in the SIT

As shown in Figure 1A, molsidomine attenuated social interaction impairment induced by ketamine. The overall analysis of the social interaction data revealed a statistically significant interaction of ketamine x molsidomine (F_2,47_ = 4.08, *p* < 0.05), a main effect of ketamine (F_1,47_ = 135.7, *p* < 0.001), and of molsidomine (F_2,47_ = 10.5, *p* < 0.001).

Post-hoc analysis comparing treatment means evidenced that ketamine + vehicle-treated rats displayed decreased social interaction levels compared to vehicle + vehicle, vehicle + molsidomine 2 mg/kg, vehicle + molsidomine 4 mg/kg, and most importantly to ketamine + molsidomine 2 mg/kg and ketamine + molsidomine 4 mg/kg-treated rats (*p* < 0.05, Figure 1A). Further, the social interaction levels expressed by rats that received ketamine + molsidomine 2 or 4 mg/kg were lower compared to that of their vehicle + molsidomine-treated cohorts (*p* < 0.05, Figure 1A).

Locomotor activity data analysis (Figure 1B) revealed a main effect of ketamine (F_1,47_ = 226.8, *p* < 0.001) of molsidomine (F_2,47_ = 3.9, *p* = 0.027) but not a significant interaction between ketamine and molsidomine (F_2,47_ = 0.68, *p* = 0.51) thus indicating that all rats treated with ketamine displayed lower motility levels as compared to their control counterparts.

### 2.2. Experiment 2: Effects of Molsidomine and Ketamine on Spatial Recognition Memory Assessed in the OLT

The analysis of the D index data revealed a significant main effect of ketamine (F_1,48_ = 25, *p* < 0.001), a significant main effect of molsidomine (F_2,47_ = 4.15, *p* = 0.023) and a significant ketamine x molsidomine interaction (F_2,47_ = 5.7, *p* = 0.06). Post-hoc comparisons showed that the ketamine + vehicle group displayed a lower discrimination index D compared to all the other experimental groups, also including the ketamine + 2 mg/kg molsidomine and ketamine + 4 mg/kg molsodomine groups (*p* < 0.05; Figure 2A).

Overall, analysis of total exploration times data expressed by different groups of rats during T2 did not show significant effects of ketamine, molsidomine, or their combination (Figure 2B).

### 2.3. Experiment 3: Effects of Sub-Threshold Doses of Molsidomine and Clozapine on Natural Forgetting Assessed in the ORT

The analysis of the D index data revealed a significant main effect of molsidomine (F_1,31_ = 49.5, *p* < 0.001), a significant main effect of clozapine (F_1,31_ = 31.1, *p* < 0.001) and a significant molsidomine x clozapine interaction (F_1,31_ = 39.4, *p* < 0.001). Post-hoc comparisons indicated that animals that received molsidomine plus clozapine discriminated significantly better novel than familiar objects with respect to the other experimental groups (*p* < 0.05; Figure 3A).

Overall, analysis of total exploration times data expressed by different groups of rats during T2 did not show significant effects of molsidomine, clozapine, or their combination (Figure 3B).

### 2.4. Experiment 4: Effects of Sub-Threshold Doses of Molsidomine and Clozapine on Ketamine-Induced Non-Spatial Recognition Memory Deficits Assessed in the ORT

The analysis of the D index data evidenced a significant main effect of ketamine (F_1,31_ = 39.9, *p* < 0.001) of the combination molsidomine + clozapine (F_1,31_ = 11.9, *p* = 0.002) and a significant interaction between ketamine and the combination treatment (F_1,31_ = 11.2, *p* = 0.002). Post-hoc comparisons showed that the ketamine + vehicle + vehicle group displayed a lower discrimination index D compared to all the other experimental groups, including the ketamine + molsidomine + clozapine group (*p* < 0.05; Figure 4A).

Overall, analysis of total exploration times data expressed by different groups of rats during T2 did not show significant effects of molsidomine, clozapine, or their combination (Figure 4B).

## 3. Discussion

The sub-chronic challenge with ketamine (8 mg/kg) remarkably diminished the time spent in social interaction and locomotor activity in the rat, in line with previous findings [35,36]. These results propose that ketamine can give rise to changes in rat behavior that may contemplate social impairments, a typical feature of schizophrenia patients. These detrimental effects of ketamine are in line with human studies, in which the ability of NMDA receptor antagonists to cause certain aspects of negative symptoms of schizophrenia has been observed [3,4,5]. Acute administration of molsidomine (2 and 4 mg/kg) in a dose-independent manner decreased the social withdrawal induced by ketamine. The reversal, however, was partial, as rats that received either ketamine or molsidomine spent significantly less amount of time in social interaction with respect to their control cohorts.

It can probably be ruled out a potential sedative action of molsidomine which might influence its beneficial effect on ketamine-induced social withdrawal because motor activity levels exhibited by ketamine plus vehicle-treated rats were superimposable to those displayed by the ketamine plus molsidomine-treated animals. In addition, the failure of molsidomine to counteract the ketamine-induced hypomotility in the SIT reduced the possibility that its effect on social withdrawal caused by ketamine was related to an increase in motor activity.

Interestingly, the disrupting effect of ketamine evidenced in the SIT cannot be attributed to its potential anxiogenic action [37] since we have previously demonstrated that using the same dose and treatment schedule of ketamine did not influence rats’ performance in the light/dark test, a procedure largely used to evaluate potential anxiogenic or anxiolytic effects of chemicals in rodents [22].

A *per se* effect of molsidomine was not observed in the SIT. The present results propose that the effects exerted by ketamine and molsidomine on animals’ performance in the SIT were unrelated to the extent of motor activity. Finally, the SIT findings are consistent with our prior observations in which the NO donor SNP was found to alleviate social withdrawal produced by ketamine in rats [22].

In line with previous reports, post-training administration of ketamine (3 mg/kg) disrupted rodents’ performance in the OLT [36,38]. Post-training administration of molsidomine (2 and 4 mg/kg) dose independently counteracted this impairing effect of ketamine on rats’ spatial recognition memory abilities. The challenge with molsidomine alone did not affect the rats’ performance in the OLT. Results here reported extended prior findings in which molsidomine attenuated non-spatial recognition memory impairments produced by another NMDA receptor antagonist, MK-801, in rats [27].

The current findings are also in agreement with previous studies in which the NO donors S-nitroso-N-acetylpenicillamine (SNAP) [39], SNP [22,40] spermine NONOate and DETANONOate [41] reversed memory deficits related to glutamatergic hypofunction.

Subsequently, the efficacy of the combination of sub-threshold doses of molsidomine (1 mg/kg) with sub-effective doses of the atypical antipsychotic clozapine (0.1 mg/kg) to alleviate delay-dependent and ketamine (3 mg/kg)-induced non-spatial recognition memory deficits were tested using the ORT. Post-training administration of this combination counteracted delay-dependent recognition memory deficits. On the contrary, the inactive doses of molsidomine and clozapine, by themselves, did not ameliorate natural forgetting. In addition, the concomitant administration of sub-effective doses of molsidomine with sub-effective doses of clozapine was demonstrated effective in reversing non-spatial recognition memory deficits caused by ketamine in the rat.

So far, few studies have been conducted aiming to investigate the potential anti-psychotic-like effect of the joint administration of a NO donor with an atypical neuroleptic. In this context, it has been shown that concomitant administration of SNP with olanzapine or risperidone but not clozapine increased rats’ performance in the conditioned avoidance response test (CARS), a behavioral procedure resembling the positive symptoms of schizophrenia patients [31,32]. The apparent failure of the combination of SNP with clozapine to induce an antipsychotic-like effect has been observed following repeated treatment, and the differential effects between olanzapine and clozapine might reside in their different mechanisms of action [31,32]. Interestingly, tolerance has been developed following clozapine’s chronic administration, and this latter finding might be ascribed to neuroplasticity variations caused by the antagonistic action of clozapine on the 5-HT_2C_ receptor located in the PFC [42].

Finally, the concomitant administration of sub-effective doses of SNP and clozapine failed to reduce attentional deficits related to glutamatergic dysfunction, evidenced in the prepulse inhibition (PPI) test [33]. Our results are partially inconsistent with this latter report. This discrepancy might be ascribed to differences in experimental settings, including the type of the animal, the preclinical model of schizophrenia investigated, the chemicals and the dose regimen used.

For the first time to our knowledge, a joint administration of sub-effective doses of an NO donor and of atypical neuroleptic alleviated recognition memory deficits, a cognitive domain impaired in schizophrenia patients [28,29]. The present findings propose a synergistic effect of the combined treatment and corroborate additional research in order to verify the clinical utility of molsidomine as an add-on treatment to neuroleptic compounds in schizophrenia.

ORT and OLT are non-spatial and spatial recognition memory paradigms, respectively, that are based on spontaneous exploratory behavior in rodents [30,34]. These paradigms do not presuppose clear reward or punishment but depend on the natural curiosity of the animals and their preference for novelty which does not seem to be determined by reinforcement/response contingencies [43]. Further, these behavioral procedures are similar to protocols used in human studies and express a consistent level of predictive validity [34].

Due to the treatment schedules, the effects expressed by the different compounds in both the recognition memory tasks (OLT and ORT) might indicate a potential modulation of post-training memory stages (consolidation and/or retrieval of information). In the experiments in which a short interval (1 h) was applied, it is difficult to separate the effects of drugs on consolidation from possible effects on retrieval. Further, due to this brief intertrial interval duration, the procedures used can be considered as short-term memory tests.

By contrast, in the ORT study, in which the effects of the joint treatment were evaluated on natural forgetting, a long interval (24 h) was used. Since the relative half-lives of molsidomine and clozapine detected in the rat tissue are 2 and 6 h, respectively [25,44], it can be concluded that the effects of compounds were evidenced in a protocol assessing long-term memory. Based on the above, this combined treatment seems to act specifically on the consolidation memory process.

In the recognition memory experiments, chemicals were injected peripherally. It cannot be excluded, therefore, that nonspecific factors (e.g., sensorimotor or motivational) might have affected rats’ cognitive performance. It has been observed, however, that the exploratory levels displayed by animals during the retention phase in all recognition memory tasks did not vary between the different experimental groups. This pattern of results proposes that the implication of unspecific parameters in the effects of ketamine, molsidomine, and clozapine on animals’ cognitive performance can probably be ruled out.

The mechanism underlying ketamine’s psychotomimetic effects has been, at least in part, attributed to the blockade of the NMDA receptor located on γ-aminobutyric acid (GABA) interneurons, which gives rise to an increment of neural activity in various structures of the limbic system [45]. This action increases neuronal activity and evokes excessive glutamate release in the PFC and limbic regions [45,46]. The above-described inhibitory action of ketamine on the NMDA receptor compromises the functionality of the neural NO synthase (nNOS)/cyclic guanosine monophosphate (cGMP) pathway [15,47,48].

The mechanism(s) of action by which molsidomine attenuates behavioral deficits caused by ketamine is not yet clarified. It has been revealed that the administration of molsidomine in nNOS KO mice reduced their memory deficits, evidenced by a contextual fear conditioning test and normalized cGMP levels [49]. In a series of studies, the restoring effect exerted by SNP on the NMDA/nNOS/CGMP signaling pathway, which functionality is impaired in schizophrenia, has been evidenced. Specifically, it has been shown that SNP attenuated hypermotility induced by PCP and normalized c-fos expression, a metabolic parameter of neuronal activity [50]. Further, this NO donor enhanced a feedback inhibition of NOS [15] and counteracted the abnormal expression of hippocampal neurons containing NOS following a challenge with ketamine [47]. Finally, SNP, by facilitating cerebral perfusion, can relieve cerebral hypoperfusion, a typical hallmark of schizophrenics [51].

An alternative explanation of the results reported here might be linked to the well-known relations between schizophrenia and oxidative stress [52]. In agreement with this is the pro-oxidant profile expressed by ketamine [53]. Taking the above into consideration, the antioxidant properties of NO donors, comprising molsidomine, revealed in various experimental models of neurodegenerative diseases [54], may constitute a feasible explanation of molsidomine’s effects. Research is mandatory, aiming to clarify this important issue.

The potential mechanism(s) through which the combined treatment exerts its effects on recognition memory is still unclear. With regard to this issue, it has been reported that atypical antipsychotics, including clozapine, stimulate DA output in the medial PFC (mPFC) and hippocampus in rats [55,56]. It appears that this stimulatory action of the atypical neuroleptics might underlie their beneficial effects on cognition deficits evidenced in schizophrenia [57]. Interestingly, co-administration of the NO donor SNP with the atypical antipsychotic risperidone potentiated its antipsychotic-like effect by enhancing the risperidone-mediated DA outflow in the mPFC, a brain area critically involved in cognition [31]. This might be a possible hypothesis to explain the efficacy of the combined treatment on recognition memory deficits.

An alternative explanation of the joint treatment findings might be due to the stimulatory action exerted by molsidomine and clozapine on the cholinergic system. Either molsidomine [58] or clozapine [56,59] was found to increase acetylcholine (Ach) release in different brain structures. Ach is a neurotransmitter critically involved in cognition. Further, the antioxidant properties of molsidomine [54] and clozapine [60] and their ability to enhance synaptic plasticity by promoting long-term potentiation (LTP) [61,62] might also be considered for the interpretation of the combined treatment results. Further research will be required to resolve this issue.

The present study presents some limitations. The behavioral effects of molsidomine and its combination with clozapine were assessed in preclinical models of schizophrenia, mimicking exclusively glutamatergic hypofunction. Molsidomine’s presumed pro-cognitive action was evaluated using only recognition memory paradigms. Further, molsidomine’s effects were assessed entirely in young male rats following acute administration. Supplementary research should be carried out in old and female rodents utilizing a variety of schizophrenia (neurodevelopmental and pharmacological) models and treatment schedules (long-term treatment), aiming to expand and validate the results presented here. Molecular, neurochemical, and electrophysiological experiments should also be designed and conducted to elucidate the potential mechanism(s) of action underlying molsidomine’s antipsychotic-like effects evidenced in behavioral studies. Involvement in these future studies of various NO modulators and atypical antipsychotics might be important.

## 4. Materials and Methods

### 4.1. Animals

Different populations of male (3-month-old) Wistar rats (Hellenic Pasteur Institute, Athens, Greece) weighing 250–300 g were used for each experiment. The rats were housed in Makrolon cages (47.5 cm length × 20.5 cm height × 27 cm width), with three per cage, in a standard environment (21 ± 1 °C; 50–55% relative humidity; 12 h/12 h light/dark cycle, lights on at 7 a.m.) with access to food and water ad libitum.

The experiments that involved animals and their care were conducted in accordance with international guidelines and national (Animal Act, P.D. 160/91) and international laws and policies (EEC Council Directive 86/609, JL 358, 1, 12 December 1987). The present study was approved by the local committee (Prefecture of Larissa, Greece, protocol number 58379/13 February 2023).

### 4.2. Behavior

#### 4.2.1. Social Interaction Test (SIT)

SIT was assessed in an activity cage (catalog number 7420, Ugo Basile, Varese, Italy). The experimental chamber consisted of a box made of Plexiglas (50 cm length × 33 cm height × 50 cm width). SIT was performed as described previously [35,36]. The rats subjected to the social interaction experiment were tested only once. All rats within a given cage received identical treatment. The rats received daily injections in their home cage for 3 days of a given compound or vehicle and were tested on the day of the last injection after the appropriate pretreatment time had expired. The 3-day treatment schedule for ketamine was selected based on studies showing that tolerance to the ataxic effects of ketamine, but not to the social behavior, rapidly develops after repeated administration [35].

On the test day, rats that had received identical treatment and were unfamiliar to each other (housed in different cages) were placed simultaneously into the apparatus in two opposite corners. The difference in body weight between the paired rats was within 10 g. Their behavior was observed for 10 min, and thereafter, they were returned to the home cage.

Social behavior for each pair member was measured. Social behaviors, scored as duration of social interaction, included: sniffing, grooming, following, kicking, mounting, jumping on, wrestling/boxing, and crawling under/over the partner. The amount of time spent by each rat in a pair in above-described behaviors was summed to produce a single social interaction score. In addition, locomotor activity expressed as the total number of counts for each pair of rats during the 10-min observation period was recorded.

#### 4.2.2. Object Location Task (OLT)

The OLT is a procedure for evaluating spatial recognition memory. This task assesses the ability of rodents to discriminate the novelty of the object locations but not the objects themselves because the testing arena is already familiar to the animals [30]. The test apparatus consisted of a dark open box made of Plexiglas (80 cm length × 50 cm height × 60 cm width) that was illuminated by a 60 W light suspended 60 cm above the box. The light intensity was equal in the different parts of the apparatus. The apparatus was placed in a large observation room and was surrounded by large external and typical objects (cues) to assist the animals in successfully performing the test. These cues were kept in a fixed position for the entire testing period. The objects were made of glass, plastic or metal and were in three different shapes—metallic cubes, glass pyramids and plastic cylinders 7 cm high—and could not be displaced by rats.

OLT was performed as described elsewhere [30,34]. Briefly, during the week before undertaking testing, the animals were handled twice daily for 3 consecutive days. Before testing, the rats were allowed to explore the empty apparatus for 2 min for 3 consecutive days. During testing, a session that consisted of two 2-min trials was carried out. During the “sample” trial T1, two identical samples (objects) were positioned in two opposite corners of the same side of the apparatus in a casual fashion, 10 cm away from the sidewalls. A rat was gently positioned in the center of the arena and allowed to inspect the two similar objects. After the sample phase T1, the rat returned to its home cage, and an intertrial interval (ITI) followed. Subsequently, the “choice” trial T2 was conducted. During T2, one of the two similar objects was moved to a different location (new location) (NL) while the other object remained in the same position (FL) as in T1. Therefore, the two objects were now in diagonal corners.

All the combinations and locations of the objects were counterbalanced to reduce the potential bias due to preferences for specific places or objects.

Exploration was defined as follows: directing the nose toward the object at a distance of 2 cm or less and/or touching the object with the nose. Turning around or sitting on the object was not considered exploratory behavior. The time spent by the rats exploring each object during T1 and T2 was manually recorded with a stopwatch. The discrimination between the FL and NL during T2 was measured by comparing the time spent exploring the object in the FL with the time spent exploring the object in the NL. Because the exploratory time may be influenced by differences in the total exploratory activity, a discrimination index (D) representing the preference for the new as opposed to familiar object position was calculated as follows: D = (NL − FL)/(NL + FL), whereas NL is the exploration time of the object in the novel location, FL that of the object in the familiar location and NL + FL is the total exploration time of both objects during T2 [63]. Correct recognition is shown by rats consistently spending more time inspecting the novel location of the object than the familiar one during the choice trial T2 [30].

#### 4.2.3. Object Recognition Task (ORT)

ORT assesses non-spatial recognition memory abilities in rodents [34]. The test apparatus was the same chamber as the one utilized in the OLT. The objects to be discriminated (in triplicate) were the same objects as in the OLT.

ORT was performed as described previously [36]. Briefly, during the week before undertaking the testing, the animals were handled twice a day for 3 consecutive days. Before testing, the rats were allowed to explore the empty apparatus for 2 min for 3 consecutive days. During testing, a session that consisted of two 2-min trials was conducted. During the “sample” trial (T1), two identical samples (objects) were positioned in two opposite corners of the apparatus in a casual fashion, 10 cm away from the sidewalls. A rat was gently positioned in the center of the arena and allowed to inspect the two similar objects. After the sample phase (T1), the rat went back to its home cage, and an intertrial interval (ITI) followed. Subsequently, the “choice” trial (T2) was conducted. During T2, a novel object substituted one of the objects presented during T1. The animals, thus, were re-exposed to two objects: a copy of the familiar F object and the novel N object.

All the combinations and positions of the objects were counterbalanced to reduce the potential bias due to preferences for specific places or objects. The definition of exploration is provided above in the context of describing the object location procedure.

The total time spent by the rats exploring the two identical objects, F1 and F2, during the sample phase T1 and the total time spent exploring the two different objects (F and N) during the choice trial T2 were manually recorded by using a stopwatch. The discrimination between F and N during T2 was measured by comparing the time spent exploring the familiar object with the time spent exploring the novel object. Because the exploratory time may be influenced by differences in the total exploratory activity, a discrimination index D representing the preference for the new as opposed to familiar object was calculated as follows: D = (N − F)/(N + F), where N is the exploration time for the novel object, F is that for the familiar object and N + F is the total exploration time for both objects during T2 [63]. Correct recognition was shown by rats consistently spending more time inspecting a novel object than the familiar one during T2 [34].

In all the above-described experiments, to avoid the presence of olfactory cues, the apparatuses and objects (where necessary) were carefully cleaned with 20% ethanol after each trial and then wiped with dry paper.

### 4.3. Drugs

Ketamine hydrochloride and molsidomine (Sigma, St. Louis, MO, USA) were dissolved in saline (NaCl 0.9%). Clozapine (Sigma, St. Louis, MO, USA) was dissolved in a minimum volume of acetic acid, made up to a volume with saline and pH adjusted to 7 with 0.1 M NaOH. All drug solutions were freshly prepared on the day of testing and were administered intraperitoneally (i.p.) in a volume of 1 mL/kg. For all studies, control animals received isovolumetric amounts of the specific vehicle solutions.

### 4.4. Experimental Protocol

Daily testing was conducted between 10 a.m. and 3 p.m. during the light phase of the light/dark cycle. Animal’s behavior was video recorded. Data evaluation was carried out by an experimenter who was unaware of the pharmacological treatment. Animals were divided into various experimental (treatment) groups according to the Completely Randomized Design (CRD).

#### 4.4.1. Experiment 1: Effects of Molsidomine on Ketamine-Induced Social Withdrawal Assessed in the SIT

The 3-day treatment schedule and the dose of ketamine (8 mg/kg) were chosen based on a prior study showing that this sub-chronic ketamine treatment caused social interaction deficit but not hyperactivity or ataxia [35]. The doses of molsidomine (2 and 4 mg/kg) were selected on the basis of a previous report [27] and on our unpublished observations.

Compounds were administered as described previously [33]. Each rat received ketamine or vehicle for two consecutive days. On day 3, 30 min before testing, rats were injected with vehicle or ketamine (8 mg/kg), and immediately after (5–10 s.), they were administered with molsidomine (2 or 4 mg/kg). 

Animals were randomly divided into six experimental groups (8 pairs of rats per group) as follows: vehicle + vehicle; vehicle + molsidomine (2 mg/kg); vehicle + molsidomine (4 mg/kg); vehicle + ketamine (8 mg/kg); ketamine (8 mg/kg) + molsidomine (2 mg/kg); ketamine (8 mg/kg) + molsidomine (4 mg/kg).

#### 4.4.2. Experiment 2: Effects of Molsidomine and Ketamine on Spatial Recognition Memory Assessed in the OLT

Appropriate treatment was performed immediately after the training (sample) trial T1. Molsidomine was administered 5–10 s after the vehicle or ketamine.

In this study, the 1-h ITI was selected since, at this delay condition, spatial recognition memory abilities are still intact in the vehicle-treated rat [36], and deficits associated with treatment with ketamine (hypermotility, stereotypies, ataxia) [8] were not observed at this time point [38]. The doses of molsidomine (2 and 4 mg/kg) were selected on the basis of a previous report [27] and on our unpublished observations. The dose of ketamine (3 mg/kg) was selected based on a previous report in which it was found to impair rats’ performance in the OLT without producing side effects [38].

Animals were randomly divided into six experimental groups with 8 rats per group as follows: vehicle + vehicle; vehicle + molsidomine (2 mg/kg); vehicle + molsidomine (4 mg/kg); vehicle + ketamine (3 mg/kg); ketamine (3 mg/kg) + molsidomine (2 mg/kg); ketamine (3 mg/kg) + molsidomine (4 mg/kg).

#### 4.4.3. Experiment 3: Effects of Sub-Threshold Doses of Molsidomine and Clozapine on Natural Forgetting Assessed in the ORT

Compounds were injected just after the sample trial T1. Clozapine was administered 5–10 s after the vehicle or molsidomine. The sub-effective doses of molsidomine (1 mg/kg) and clozapine (0.1 mg/kg) were selected on the basis of previous studies [64,65].

In this experiment, the ITI at which recognition memory disappeared in the normal rat was used. Thus, a 24 h retention was utilized [66].

Animals were randomly divided into four experimental groups with 8 rats per group as follows: vehicle + vehicle; vehicle + molsidomine (1 mg/kg); vehicle + clozapine (0.1 mg/kg); molsidomine (1 mg/kg) + clozapine (0.1 mg/kg).

#### 4.4.4. Experiment 4: Effects of Sub-Threshold Doses of Molsidomine and Clozapine on Ketamine-Induced Non-Spatial Recognition Memory Deficits Assessed in the ORT

Compounds were injected just after the sample trial T1. Clozapine was administered 5–10 s after the vehicle or molsidomine. The sub-effective doses of molsidomine (1 mg/kg) and clozapine (0.1 mg/kg) were selected on the basis of previous studies [65,66]. In this study, the 1-h ITI was selected since, at this delay, condition non-spatial recognition memory abilities are still intact in the vehicle-treated rat [36], and deficits associated with treatment with ketamine (hypermotility, stereotypies, ataxia) [8] were not observed at this time point [38].

Animals were randomly divided into four experimental groups with 8 rats per group as follows: vehicle + vehicle + vehicle; vehicle + molsidomine (1 mg/kg) + clozapine (0.1 mg/kg); vehicle + vehicle + ketamine (3 mg/kg); ketamine (3 mg/kg) + molsidomine (1 mg/kg) + clozapine (0.1 mg/kg).

### 4.5. Statistical Analysis

Experiments data were expressed as mean ± S.E.M. and were analyzed using two-way analysis of variance (ANOVA). Post-hoc comparisons between treatment means were made using the Tukey’s *t*-test, but only when a significant interaction between ketamine and molsidomine (experiments 1 and 2) or molsidomine and clozapine (experiment 3) or ketamine and the combination treatment clozapine + molsidomine (experiment 4) was achieved. Values of *p* < 0.05 were considered statistically significant [67].

## 5. Conclusions

Results reported here indicate that the NO donor molsidomine reversed behavioral deficits related to the blockade of the NMDA receptor. These findings might have a translational value since these effects of molsidomine were revealed in preclinical procedures resembling negative symptoms and cognitive deficits, typical features the schizophrenia patients. Further, molsidomine’s ability to enhance the antipsychotic-like action of the second-generation neuroleptic clozapine has been revealed. The latter suggests that the combined use of molsidomine and an atypical neuroleptic might represent a new strategy for the therapy of schizophrenia. Additional preclinical intense research is required aiming to further investigate the potential antipsychotic-like profile of NO donors and establish a definitive role for them in the therapy of schizophrenia.

## Figures and Tables

**Figure 1 ijms-24-06866-f001:**
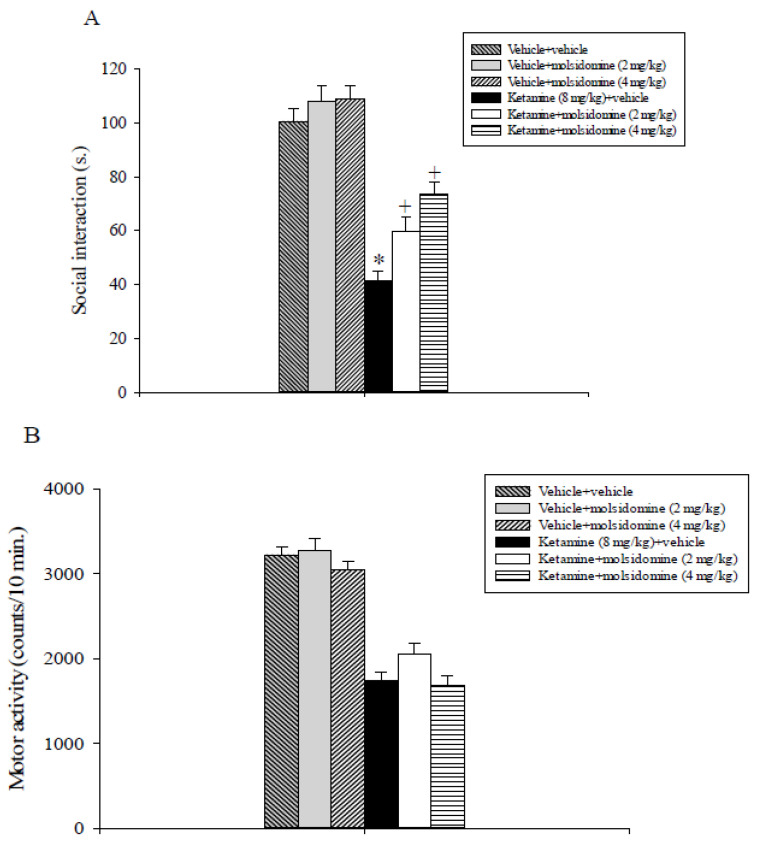
Social interaction test. The histogram represents the mean ± S.E.M of 8 pairs of rats per treatment group. (**A**) Social interaction levels expressed by different groups of rats. * *p <* 0.05 vs. all the other groups; + *p <* 0.05 vs. the vehicle + vehicle, vehicle + molsidomine 2 mg/kg and vehicle + molsidomine 4 mg/kg groups. (**B**) Locomotor activity expressed by different groups of rats during the social interaction test.

**Figure 2 ijms-24-06866-f002:**
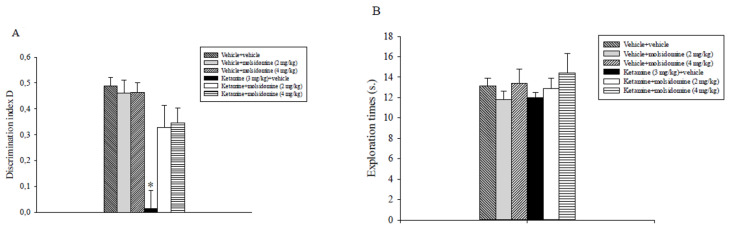
Object location task. The histogram represents the mean ± S.E.M of 8 rats per treatment group. The 1-h ITI was used. (**A**) Discrimination index D performance expressed by different groups of rats during T2. * *p* < 0.05 vs. all the other groups. (**B**) Total exploration times.

**Figure 3 ijms-24-06866-f003:**
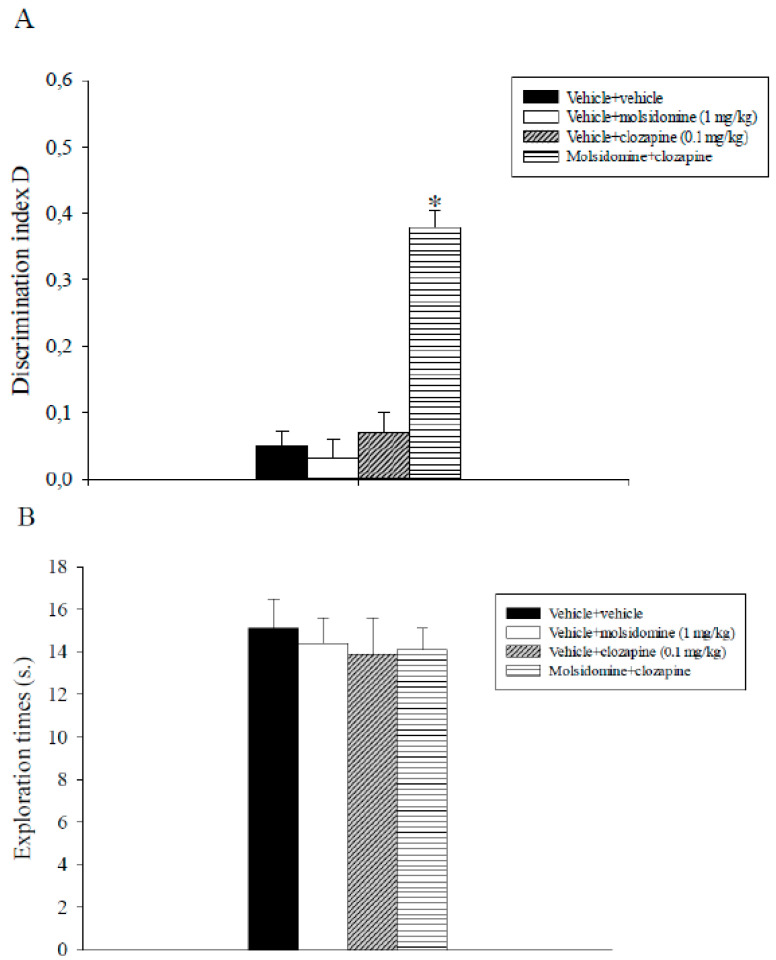
Object recognition task. The histograms represent the mean ± S.E.M of 8 rats per treatment group. The 24-h ITI was used. (**A**) Discrimination index D performance expressed by different groups of rats during T2. * *p* < 0.05 vs. all the other groups. (**B**) Total exploration times.

**Figure 4 ijms-24-06866-f004:**
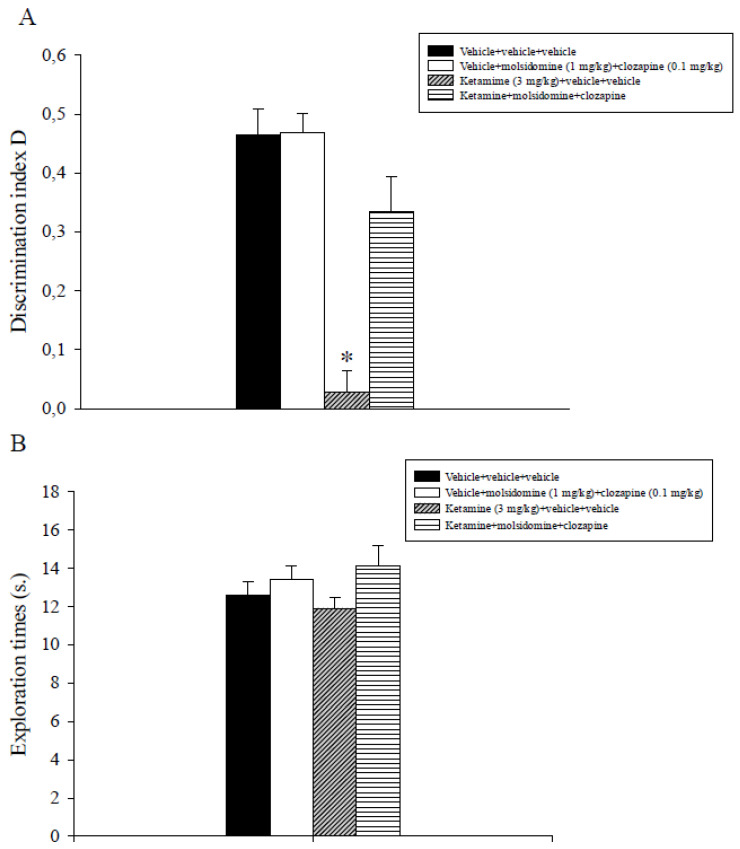
Object recognition task. The histograms represent the mean ± S.E.M of 8 rats per treatment group. The 1-h ITI was used. (**A**) Discrimination index D performance expressed by different groups of rats during T2. * *p* < 0.05 vs. all the other groups. (**B**) Total exploration times.

## Data Availability

The data presented in this study are available on request from the corresponding author.

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
