# Peer review of "The Nitric Oxide (NO) Donor Molsidomine Counteract Social Withdrawal and Cognition Deficits Induced by Blockade of the NMDA Receptor in the Rat"

_ijms, 2023, doi:10.3390/ijms24076866_

Round 1
Reviewer 1 Report
Overall, the introduction provides a comprehensive background on schizophrenia and the role of the glutamatergic system, as well as the potential therapeutic effects of nitric oxide (NO) donors. The authors also adequately present the rationale for investigating molsidomine as a potential treatment for schizophrenia. I provide the following comments and suggestions for the manuscript:
- The manuscript is well-organized and presents the results of an interesting study on the effects of ketamine and molsidomine in rats, as well as the potential benefits of a combined treatment with an atypical antipsychotic. The introduction could benefit from further elaboration on the significance of the study and the current state of knowledge in the field.
- In lines 31-32, consider rephrasing "Epidemiological evi- 31 dence suggests that up to 1% of the world population suffer from this mental disease." to "Epidemiological evidence suggests that schizophrenia affects up to 1% of the world population."
- In lines 58-59, there seems to be a punctuation error: "is 58 an important intra- and inter-cellular messenger in the brain [12]." A period should be placed after "gas" and the subsequent sentence should start with "It is an important..."
- In lines 60-61, "Underproduction of it" should be replaced with "Underproduction of NO."
- Throughout the introduction, there are several instances of inconsistent hyphenation and line breaks. Please ensure that the formatting is consistent and easy to read.
- In lines 89-100, the authors outline the study's purpose and the experimental methods used. It would be helpful to include a brief statement about the study's hypotheses, e.g., "We hypothesize that molsidomine will alleviate social isolation and spatial recognition memory deficits induced by ketamine."
- Consider rephrasing the last sentence of the introduction (lines 99-100) for clarity: "For these latter studies, the object recognition task (ORT) was utilized [31]." to "For these latter studies, we utilized the object recognition task (ORT) [31]."
- The methods section should be expanded to provide more detailed information on the experimental design, including the number of rats used in each group, the process for randomization, and any measures taken to minimize potential biases.
9. In the results section, it would be helpful to provide more detailed statistics for the reported differences between groups, such as effect sizes and confidence intervals, in addition to the p-values. The authors have presented the data clearly and supported their claims with appropriate statistical analysis.
- The discussion section is thorough and considers alternative explanations for the findings. However, the authors should address the limitations of their study more explicitly, such as the generalizability of their findings to other models of schizophrenia and the potential influence of factors like sex and age on the observed effects. Moreover, the behavioral effects of molsidomine and its combination with clozapine were assessed only in preclinical models of schizophrenia mimicking glutamatergic hypofunction and were evidenced following acute administration. Further research is needed using other schizophrenia models and treatment schedules to validate the results.
- The conclusion could be more concise and should emphasize the key findings and their implications for the field. The authors should also consider providing more specific recommendations for future research, such as the investigation of other NO donors, potential interactions with other neurotransmitter systems, and the exploration of long-term effects and therapeutic strategies.
- Throughout the manuscript, there are some minor grammatical and typographical errors that should be corrected to improve readability. For example, in the section discussing the effects of ketamine and molsidomine on rats' performance in the SIT (lines 194-198), the sentence structure could be improved for clarity.
- The authors should ensure that all references are formatted consistently and that all cited works are included in the reference list.
Overall, the study provides valuable insights into the potential benefits of combining an NO donor with an atypical antipsychotic for the treatment of cognitive deficits in schizophrenia. With some revisions and improvements, this manuscript could make a valuable contribution to the literature.
Author Response
v
Reviewer 1 comments:
- Overall, the introduction provides a comprehensive background on schizophrenia and the role of the glutamatergic system, as well as the potential therapeutic effects of nitric oxide (NO) donors. The authors also adequately present the rationale for investigating molsidomine as a potential treatment for schizophrenia. I provide the following comments and suggestions for the manuscript:
The manuscript is well-organized and presents the results of an interesting study on the effects of ketamine and molsidomine in rats, as well as the potential benefits of a combined treatment with an atypical antipsychotic. The introduction could benefit from further elaboration on the significance of the study and the current state of knowledge in the field.
Answer
Thank you for nice comments.
2. In lines 31-32, consider rephrasing "Epidemiological evidence suggests that up to 1% of the world population suffer from this mental disease." to "Epidemiological evidence suggests that schizophrenia affects up to 1% of the world population."
Answer
Corrected as kindly suggested. Please see lines 31-32.
3. In lines 58-59, there seems to be a punctuation error: "is 58 an important intra- and inter-cellular messenger in the brain [12]." A period should be placed after "gas" and the subsequent sentence should start with "It is an important..."
Answer
Corrected as kindly suggested. Please see lines 58-59.
4. In lines 60-61, "Underproduction of it" should be replaced with "Underproduction of NO."
Answer
Corrected accordingly. Please see line 60.
5. Throughout the introduction, there are several instances of inconsistent hyphenation and line breaks. Please ensure that the formatting is consistent and easy to read.
Answer
We did our best to improve formatting.
6. In lines 89-100, the authors outline the study's purpose and the experimental methods used. It would be helpful to include a brief statement about the study's hypotheses, e.g., "We hypothesize that molsidomine will alleviate social isolation and spatial recognition memory deficits induced by ketamine."
Answer
Revised as kindly requested. Please see lines 93-98.
7. Consider rephrasing the last sentence of the introduction (lines 99-100) for clarity: "For these latter studies, the object recognition task (ORT) was utilized [31]." to "For these latter studies, we utilized the object recognition task (ORT) [31]."
Answer
Sentence was revised accordingly. Please see line 107.
- The methods section should be expanded to provide more detailed information on the experimental design, including the number of rats used in each group, the process for randomization, and any measures taken to minimize potential biases.
Answer
As in our first submission, the number of rats per experimental group and the measures taken to minimize potential bias (experimenter unaware of the pharmacological treatment evaluated behavioural data) are included also in our revised manuscript. Please see paragraphs 4.4.1., 4.4.2., 4.4.3. and 4.4.4. Please see also lines 449-450.
As kindly suggested by the Reviewer the process of randomization is included in our revised text. Please see lines 450-452.
9. In the results section, it would be helpful to provide more detailed statistics for the reported differences between groups, such as effect sizes and confidence intervals, in addition to the p-values. The authors have presented the data clearly and supported their claims with appropriate statistical analysis.
Answer
We appreciate Reviewers’ positive comment regarding the appropriateness of the statistical elaboration of our data. The issue concerning additional statistical analyses was presented and discussed with our expert statistician. The statistician strongly believes that the two-way ANOVA conducted in our study, in which not only P values but also F ratios of the main effects and interactions are reported, represent the most appropriate statistical analyses of our data. He suggested, therefore, to not add further statistical elaborations.
10. The discussion section is thorough and considers alternative explanations for the findings. However, the authors should address the limitations of their study more explicitly, such as the generalizability of their findings to other models of schizophrenia and the potential influence of factors like sex and age on the observed effects. Moreover, the behavioral effects of molsidomine and its combination with clozapine were assessed only in preclinical models of schizophrenia mimicking glutamatergic hypofunction and were evidenced following acute administration. Further research is needed using other schizophrenia models and treatment schedules to validate the results.
Answer
We revised the issue accordingly to Reviewer suggestions. Please see lines 320-332.
11. The conclusion could be more concise and should emphasize the key findings and their implications for the field. The authors should also consider providing more specific recommendations for future research, such as the investigation of other NO donors, potential interactions with other neurotransmitter systems, and the exploration of long-term effects and therapeutic strategies.
Answer
We did our best to address this issue. Please see lines 513-522 and also lines 320-332.
12. Throughout the manuscript, there are some minor grammatical and typographical errors that should be corrected to improve readability. For example, in the section discussing the effects of ketamine and molsidomine on rats' performance in the SIT (lines 194-198), the sentence structure could be improved for clarity.
Answer
Section revised accordingly. Please see lines 201-205.
13. The authors should ensure that all references are formatted consistently and that all cited works are included in the reference list.
Answer
We did our best to address the relative issue.

Reviewer 2 Report
This paper describes the in vivo testing of the molsidomine effect on the schizophrenia-like impairments induced by ketamine in rats. Schizophrenia is a chronic mental disorder affecting approximately 1% of the global population. The diagnostic criteria of the disease include three main groups of symptoms: positive symptoms, negative symptoms and cognitive impairment. It is estimated that negative symptoms (e.g. impaired speech, deprivation of emotion and pleasure feeling) affect 40% of patients, and marked cognitive impairment (e.g. verbal and working memory) covers up to 80%. Abnormalities in the regulation of glutamatergic transmission have been shown to contribute indirectly to positive and negative symptoms and cognitive dysfunction. Previously, it was reported that repeated challenge with ketamine reduced the time spent in social contacts in rats and impaired memory processes. Thus, the Authors chose a good reliable model for the hypothesis. Of particular interest and utility is the ability of the molsidomine to antagonize ketamine-induced memory deficits and social withdrawal. Overall it is a well-designed study and provides very interesting results. The results can be of relevance for the study and identification of potential new pharmacotherapy in schizophrenia. However, some issues raised my concerns, which I have listed below:
1. I suggest to correct the titles of the experiments. They seem to be a repetition of a few words – “effect”, “antagonizing” and “induced”, they are not fully understood.
2. I have some doubts about a figure 1 caption - why the p<0.05 value marked with "+" is not compared to the vehicle + ketamine group?
3. How locomotor activity was measured in the OLT?
4. Could you explain why the vehicle group in ORT (Figure 2A) was compared to the molsidomine+clozapine group? Conversely comparison seems to be more accurate.
5. The effects of molsidomine on ketamine induced memory impairment in the ORT was not statistically significant? (Fig. 4A)
Why did you use two similar animal tests to measure memory processes?
Author Response
Reviewer 2 comments:
This paper describes the in vivo testing of the molsidomine effect on the schizophrenia-like impairments induced by ketamine in rats. Schizophrenia is a chronic mental disorder affecting approximately 1% of the global population. The diagnostic criteria of the disease include three main groups of symptoms: positive symptoms, negative symptoms and cognitive impairment. It is
estimated that negative symptoms (e.g. impaired speech, deprivation of emotion and pleasure feeling) affect 40% of patients, and marked cognitive impairment (e.g. verbal and working memory) covers up to 80%. Abnormalities in the regulation of glutamatergic transmission have been shown to contribute indirectly to positive and negative symptoms and cognitive dysfunction. Previously, it was reported that repeated challenge with ketamine reduced the time spent in social contacts in rats and impaired memory processes. Thus, the Authors chose a good reliable model for the hypothesis. Of particular interest and utility is the ability of the molsidomine to antagonize ketamine-induced memory deficits and social withdrawal. Overall it is a well-designed study and provides very interesting results. The results can be of relevance for the study and identification of potential new pharmacotherapy in schizophrenia. However, some issues raised my concerns, which I have listed below:
Answer
We appreciate Reviewer’s’ comments.
1. I suggest to correct the titles of the experiments. They seem to be a repetition of a few words – “effect”, “antagonizing” and “induced”, they are not fully understood.
Answer
We did our best regarding this issue. Please see “Results” and “Experimental Protocol”.
2. I have some doubts about a figure 1 caption - why the p<0.05 value marked with "+" is not compared to the vehicle + ketamine group?
Answer
Figures’ 1 caption has been corrected.
- How locomotor activity was measured in the OLT?
Answer
Locomotor activity was not recorded in both recognition memory paradigms (OLT and ORT). The parameter which takes into account eventual sensorimotor, attentional or motivation issues related to the drug treatment is the total exploration time of objects. Data relative to this highly important parameter were statistically analyzed and extensively commented throughout our manuscript. Please see lines 269-275.
4. Could you explain why the vehicle group in ORT (Figure 2A) was compared to the molsidomine+clozapine group? Conversely comparison seems to be more accurate.
Answer
Probably you are referring to Fig. 3A. Relative caption was corrected as kindly suggested.
5. The effects of molsidomine on ketamine induced memory impairment in the ORT was not statistically significant? (Fig. 4A).
Answer
Relative caption was corrected as kindly suggested.
6. Why did you use two similar animal tests to measure memory processes?
Answer
OLT and ORT examine rodents’ recognition memory abilities. There are some differences however, between OLT and ORT. OLT assesses spatial memory while ORT non-spatial memory. Both tasks have a good translational value since recognition memory is impaired to schizophrenia patients. In this context, the utilization of other behavioural paradigms (mazes, avoidance procedures) for assessing cognition might be matter of future studies. Technical problems did not allow us to utilize the step-through passive avoidance test to this end. A comment regarding these limitations of our study has been added in our revised manuscript. Please see lines 320-332.
Reviewer 2 comments:
This paper describes the in vivo testing of the molsidomine effect on the schizophrenia-like impairments induced by ketamine in rats. Schizophrenia is a chronic mental disorder affecting approximately 1% of the global population. The diagnostic criteria of the disease include three main groups of symptoms: positive symptoms, negative symptoms and cognitive impairment. It is
estimated that negative symptoms (e.g. impaired speech, deprivation of emotion and pleasure feeling) affect 40% of patients, and marked cognitive impairment (e.g. verbal and working memory) covers up to 80%. Abnormalities in the regulation of glutamatergic transmission have been shown to contribute indirectly to positive and negative symptoms and cognitive dysfunction. Previously, it was reported that repeated challenge with ketamine reduced the time spent in social contacts in rats and impaired memory processes. Thus, the Authors chose a good reliable model for the hypothesis. Of particular interest and utility is the ability of the molsidomine to antagonize ketamine-induced memory deficits and social withdrawal. Overall it is a well-designed study and provides very interesting results. The results can be of relevance for the study and identification of potential new pharmacotherapy in schizophrenia. However, some issues raised my concerns, which I have listed below:
Answer
We appreciate Reviewer’s’ comments.
1. I suggest to correct the titles of the experiments. They seem to be a repetition of a few words – “effect”, “antagonizing” and “induced”, they are not fully understood.
Answer
We did our best regarding this issue. Please see “Results” and “Experimental Protocol”.
2. I have some doubts about a figure 1 caption - why the p<0.05 value marked with "+" is not compared to the vehicle + ketamine group?
Answer
Figures’ 1 caption has been corrected.
- How locomotor activity was measured in the OLT?
Answer
Locomotor activity was not recorded in both recognition memory paradigms (OLT and ORT). The parameter which takes into account eventual sensorimotor, attentional or motivation issues related to the drug treatment is the total exploration time of objects. Data relative to this highly important parameter were statistically analyzed and extensively commented throughout our manuscript. Please see lines 269-275.
4. Could you explain why the vehicle group in ORT (Figure 2A) was compared to the molsidomine+clozapine group? Conversely comparison seems to be more accurate.
Answer
Probably you are referring to Fig. 3A. Relative caption was corrected as kindly suggested.
5. The effects of molsidomine on ketamine induced memory impairment in the ORT was not statistically significant? (Fig. 4A).
Answer
Relative caption was corrected as kindly suggested.
6. Why did you use two similar animal tests to measure memory processes?
Answer
OLT and ORT examine rodents’ recognition memory abilities. There are some differences however, between OLT and ORT. OLT assesses spatial memory while ORT non-spatial memory. Both tasks have a good translational value since recognition memory is impaired to schizophrenia patients. In this context, the utilization of other behavioural paradigms (mazes, avoidance procedures) for assessing cognition might be matter of future studies. Technical problems did not allow us to utilize the step-through passive avoidance test to this end. A comment regarding these limitations of our study has been added in our revised manuscript. Please see lines 320-332.
Reviewer 2 comments:
This paper describes the in vivo testing of the molsidomine effect on the schizophrenia-like impairments induced by ketamine in rats. Schizophrenia is a chronic mental disorder affecting approximately 1% of the global population. The diagnostic criteria of the disease include three main groups of symptoms: positive symptoms, negative symptoms and cognitive impairment. It is
estimated that negative symptoms (e.g. impaired speech, deprivation of emotion and pleasure feeling) affect 40% of patients, and marked cognitive impairment (e.g. verbal and working memory) covers up to 80%. Abnormalities in the regulation of glutamatergic transmission have been shown to contribute indirectly to positive and negative symptoms and cognitive dysfunction. Previously, it was reported that repeated challenge with ketamine reduced the time spent in social contacts in rats and impaired memory processes. Thus, the Authors chose a good reliable model for the hypothesis. Of particular interest and utility is the ability of the molsidomine to antagonize ketamine-induced memory deficits and social withdrawal. Overall it is a well-designed study and provides very interesting results. The results can be of relevance for the study and identification of potential new pharmacotherapy in schizophrenia. However, some issues raised my concerns, which I have listed below:
Answer
We appreciate Reviewer’s’ comments.
1. I suggest to correct the titles of the experiments. They seem to be a repetition of a few words – “effect”, “antagonizing” and “induced”, they are not fully understood.
Answer
We did our best regarding this issue. Please see “Results” and “Experimental Protocol”.
2. I have some doubts about a figure 1 caption - why the p<0.05 value marked with "+" is not compared to the vehicle + ketamine group?
Answer
Figures’ 1 caption has been corrected.
- How locomotor activity was measured in the OLT?
Answer
Locomotor activity was not recorded in both recognition memory paradigms (OLT and ORT). The parameter which takes into account eventual sensorimotor, attentional or motivation issues related to the drug treatment is the total exploration time of objects. Data relative to this highly important parameter were statistically analyzed and extensively commented throughout our manuscript. Please see lines 269-275.
4. Could you explain why the vehicle group in ORT (Figure 2A) was compared to the molsidomine+clozapine group? Conversely comparison seems to be more accurate.
Answer
Probably you are referring to Fig. 3A. Relative caption was corrected as kindly suggested.
5. The effects of molsidomine on ketamine induced memory impairment in the ORT was not statistically significant? (Fig. 4A).
Answer
Relative caption was corrected as kindly suggested.
6. Why did you use two similar animal tests to measure memory processes?
Answer
OLT and ORT examine rodents’ recognition memory abilities. There are some differences however, between OLT and ORT. OLT assesses spatial memory while ORT non-spatial memory. Both tasks have a good translational value since recognition memory is impaired to schizophrenia patients. In this context, the utilization of other behavioural paradigms (mazes, avoidance procedures) for assessing cognition might be matter of future studies. Technical problems did not allow us to utilize the step-through passive avoidance test to this end. A comment regarding these limitations of our study has been added in our revised manuscript. Please see lines 320-332.

Reviewer 3 Report
Katsanou et al tested whether the nitric oxide (NO) donor Molsidomine could rescue behavioral deficits induced by the NMDA receptor antagonist ketamine in rats. Molsidomine (2 and 4 mg/ml) ameliorated ketamine-induced social interaction and object location deficits. Subeffective dose of Molsidomine (1 mg/kg) combined with subeffective dose of Clozapine (0.1 mg/kg) also ameliorated ketamine-induced object location impairment and enabled naïve mice to perform above chance level in a novel object recognition test with intertrial interval of 24h. Katsanou et al suggest Molsidomine as an add-on treatment that would enable to lower doses of Clozapine to reduce side effects.
As described in the discussion, the same group has previously shown that another NO donor, sodium nitroprusside (SNP), reduced ketamine-induced social withdrawal and that Molsidomine attenuated MK801-induced recognition memory impairments. Also, there was no attempt to start looking into possible mechanisms of the action.
On the positive side: Katsanou et al showed total locomotion as control for behavioral tests, the methods provide rational for the selected conditions and are detailed. Since readers do not necessarily check the methods section, moving some of the information to the results section before the results of each experiment would make the paper clearer.
1) It is preferable to avoid the term “schizophrenic” (https://www.psychiatry.org/news-room/reporting-on-mental-health-conditions
2) “The implication of NO in schizophrenia is clearly documented” might be an overstatement? At the minimum, add some strong references.
3) The two doses of NO (2 and 4 mg/ml) have very similar effects, not providing information on dose dependence.
4) Any idea why Ketamine reduced locomotion in social interaction test (Fig 1B) but not in novel object location test (Fig 2B)?
5) Experiment/Fig 4: no group of Ketamine + vehicle+ clozapine, so cannot know whether the rescue effect is due to Molsidomine, clozapine or their combination.
6) Line 235: “propose a functional interaction between clozapine and the nitrergic system: in Fig 3 the effect of Molsidomine + Clozapine is definitely synergistic rather than additive, but does that necessarily mean that there is a functional interaction? Couldn’t each drug improve memory subeffectively via a different pathway leading to a significant result?
7) Methods: How long were T1 and T2 in social interaction test and object location test?
Minor:
8) Line 144: better to use “novel” and “familiar” rather than “N” and “F”, which are only defined on line 409
9) Line 226: PFC was already defined on line 43
10) Line 366 vs. line 372-373 unclear. I guess that during the sample trial T1 the objects were placed in opposite corners of the same side of the box.
Author Response
x
Reviewer 3 comments:
Katsanou et al tested whether the nitric oxide (NO) donor Molsidomine could rescue behavioral deficits induced by the NMDA receptor antagonist ketamine in rats. Molsidomine (2 and 4 mg/ml) ameliorated ketamine-induced social interaction and object location deficits. Subeffective dose of Molsidomine (1 mg/kg) combined with subeffective dose of Clozapine (0.1 mg/kg) also ameliorated ketamine-induced object location impairment and enabled naïve mice to perform above chance level in a novel object recognition test with intertrial interval of 24h. Katsanou et al suggest Molsidomine as an add-on treatment that would enable to lower doses of Clozapine to reduce side effects.As described in the discussion, the same group has previously shown that another NO donor, sodium nitroprusside (SNP), reduced ketamine-induced social withdrawal and that Molsidomine attenuated MK801-induced recognition memory impairments. Also, there was no attempt to start looking into possible mechanisms of the action.
Answer
We fully agree with Reviewer that additional research is required to fully clarify the exact role of molsidomine in schizophrenia. Nonetheless, a part of Discussion is dedicated to the description of potential mechanisms of action underlying the effects of compounds. Please see lines 283-319. Further, the limitations of the present study are acknowledged in Discussion. Finally, a brief description of some future preclinical studies is also provided in our revised text. Please see lines 320-332.
On the positive side: Katsanou et al showed total locomotion as control for behavioral tests, the methods provide rational for the selected conditions and are detailed. Since readers do not necessarily check the methods section, moving some of the information to the results section before the results of each experiment would make the paper clearer.
Answer
We thank Reviewer for his/her suggestion and nice comments regarding our study. We believe, however, that the most appropriate section of the manuscript for this information remains the “Material and Methods section”. Therefore, we prefer to do not move them from the “Material and Methods section”. We leave the decision to the Editor for this point.
1) It is preferable to avoid the term “schizophrenic” (https://www.psychiatry.org/news-room/reporting-on-mental-health-conditions).
Answer
Done as kindly suggested. The term “schizophrenic” has been replaced by the term “schizophrenia patient” throughout the revised manuscript.
2) “The implication of NO in schizophrenia is clearly documented” might be an overstatement? At the minimum, add some strong references.
Answer
We fully agree with Reviewers’ consideration. The sentence has been revised accordingly and an appropriate reference has been added. Please see lines 59-60.
3) The two doses of NO (2 and 4 mg/ml) have very similar effects, not providing information on dose dependence.
Answer
We agree with Reviewers’ observation. This is acknowledged in our revised manuscript. Please see lines 185-186 and 208.
4) Any idea why Ketamine reduced locomotion in social interaction test (Fig 1B) but not in novel object location test (Fig 2B)?
Answer
Accordingly, to the literature of social interaction [e.g., reference 35 of the present manuscript] the ketamine treatment schedule used in social interaction test (3-day sub-chronic treatment, 8 mg/kg once per day) leads to tolerance to the ataxic effect of ketamine, but not to the social behaviour and reduction of locomotor activity. Motor functions expressed by animals in the social interaction test cannot be compared to those expressed by a single animal in a motor activity cage. In the social interaction test two unfamiliar animals are introduced in the test arena while in a motor activity procedure animals are individually placed in the motor activity cage. The hypomotility evidenced in the social interaction test following sub-chronic challenge with ketamine might be a consequence of ketamine’s property to suppress general activity expressed as social contacts between two unfamiliar rats which are introduced for the first time in an unfamiliar arena.
Concerning OLT, we did not measure motor activities, but we recorded total exploration times of objects. The latter is an important parameter which takes into account eventual sensorimotor, attentional or motivation issues related to the drug treatment. Please see lines 269-275.
5) Experiment/Fig 4: no group of Ketamine + vehicle+ clozapine, so cannot know whether the rescue effect is due to Molsidomine, clozapine or their combination.
Answer
Results of experiment 3 indicate that either clozapine + vehicle or molsidomine + vehicle did not antagonize natural forgetting (delay-dependent deficits) while their combination was effective in this context. Based on the above findings, these two groups were not included in the design of experiment 4. A relative comment is now added in our revised manuscript. Please see lines 219-221.
6) Line 235: “propose a functional interaction between clozapine and the nitrergic system: in Fig 3 the effect of Molsidomine + Clozapine is definitely synergistic rather than additive, but does that necessarily mean that there is a functional interaction?
Answer
Reviewers’ reflections are interesting. We agree with his/her statement “that the joint treatment between molsidomine and clozapine produces a synergistic effect”. Relative sentence has been corrected accordingly. Please see lines 245-246.
Couldn’t each drug improve memory subeffectively via a different pathway leading to a significant result?
Answer
This is an interesting question which needs to be addressed. Potential mechanism(s) underlying the benefits of the joint treatment have been commented. Please see lines 302-319.
7) Methods: How long were T1 and T2 in social interaction test and object location test?
Answer
T1 (sample trial) and T2 (choice trial) are procedures of the OLT and ORT. Their duration is reported in our revised text. Please see lines 382 and 414.
Minor:
8) Line 144: better to use “novel” and “familiar” rather than “N” and “F”, which are only defined on line 409.
Answer
Done as kindly suggested. Please see line 152.
9) Line 226: PFC was already defined on line 43.
Answer
Corrected as kindly suggested. Please see line 236.
10) Line 366 vs. line 372-373 unclear. I guess that during the sample trial T1 the objects were placed in opposite corners of the same side of the box.
Answer
Correct. We rephrased accordingly. Please see lines 382-390.
